# A Proposed Method to Assess the Mechanical Properties of Treadmill Surfaces

**DOI:** 10.3390/s20092724

**Published:** 2020-05-10

**Authors:** Enrique Colino, Jorge Garcia-Unanue, Bas Van Hooren, Leonor Gallardo, Kenneth Meijer, Alejandro Lucia, Jose Luis Felipe

**Affiliations:** 1IGOID Research Group, Physical Activity and Sport Sciences Department, University of Castilla-La Mancha, 45071 Toledo, Spain; enrique.colino@uclm.es (E.C.); Leonor.Gallardo@uclm.es (L.G.); 2Department of Nutrition and Movement Sciences, NUTRIM School of Nutrition and Translational Research in Metabolism, Maastricht University, Universiteitssingel 50, 6229 ER Maastricht, The Netherlands; basvanhooren@hotmail.com (B.V.H.); kenneth.meijer@maastrichtuniversity.nl (K.M.); 3School of Sport Sciences, Universidad Europea de Madrid, 28670 Madrid, Spain; alejandro.lucia@universidadeuropea.es (A.L.); joseluis.felipe@universidadeuropea.es (J.L.F.); 4Research Institute Hospital 12 de Octubre (‘imas12’), 28041 Madrid, Spain

**Keywords:** sport surfaces, running, biomechanics, shock absorption, energy restitution

## Abstract

The aim of this study was to define a reliable and sensitive test method for assessing Shock Absorption (SA), Vertical Deformation (VD), and Energy Restitution (ER) in treadmill surfaces. A total of 42 treadmills belonging to four different models were included in the study: (a) Technogym Jog700 Excite (*n* = 10), (b) Technogym Artis Run (*n* = 12), (c) LifeFitness Integrity Series 97T (*n* = 11), and (d) LifeFitness Integrity Series DX (*n* = 9). An advanced artificial athlete (AAA) device was used to assess SA, VD, and ER at three different locations along the longitudinal axis of each treadmill and in the support area of the athletes’ feet. For each location, our results show that the error assumed when performing one impact with the AAA instead of three (SA ≤ |0.1|%, VD ≤ |0.0| mm, and ER ≤ |0.2|%) is lower than the smallest changes that can be detected by the measuring device (SA = 0.4%, VD = 0.2 mm, and ER = 0.9%). Also, our results show the ability of the test method to detect meaningful differences between locations once the one-impact criterium is adopted, since absolute minimum differences between zones (SA: |0.6|%, VD: |0.3| mm, and ER: |1.2|%) were above the uncertainty of the measuring device. Therefore, performing a single impact with the AAA in each of the three locations described in this study can be considered a representative and reliable method for assessing SA, VD, and ER in treadmill surfaces.

## 1. Introduction

It is has been known for decades that running mechanics can be affected by the ground surface characteristics [1]. Indeed, surface compliance—that is, the ability of the surface to deform when subjected to an applied force [2]—influences ground contact time, step length, ground reaction forces, and running speed [1]. As such, surface compliance is also a potential determinant of the energetic cost of running and ultimately, of running performance, as well as of injury risk [3,4,5].

Compliant surfaces deform during the footfall while running, thereby allowing energy to be transferred from the athlete to the ground, with part of this energy being returned back to the athlete during the subsequent propelling phase. The amount of energy stored by the surface—and, thus, the energy potentially returned to the athlete—increases with compliance [6]. Yet, compliance itself does not fully represent the dynamic behavior of the surface. For example, surfaces such as sand or trampolines, which are both very compliant and absorb a great amount of energy, behave in a very different way while running on them, the former affecting running performance due to the low percentage of energy returned [7,8], and the latest hampering the normal execution of the running action for opposite reasons. Thus, the amount of energy returned to the athlete (and the moment at which this occurs) depend not only on compliance (or stiffness) but also on the viscoelastic (or damping) properties of the surface [1], which are related to the elasticity of its components and reflect the ability of the surface to return to its original shape.

Due to the above, other mechanical properties are used for assessing the role of sport surfaces in the athlete-surface interaction. One is shock absorption (SA), which reflects the ability of the surface to absorb impact forces during foot landing, and is associated with risk of injury [8,9]. In turn, the SA of the surface is strongly related to its vertical deformation (VD), which reflects the extent to which the surface will deform under a certain load [10,11]. Finally, energy restitution (ER) reflects the amount of energy that the surface will return back to the athlete [2]. 

In order to reduce injury risk and to ensure equal conditions for all athletes, the mechanical properties of sports surfaces must be regularly assessed and controlled according to the regulations of the relevant international sport federations (i.e., International Federation of Football Association (FIFA) and World Athletics (former International Association of Athletics Federations, IAAF), or the European Committee for Standardization (CEN), among others. Thus, at least two standards usually exist for the same surface, as in the case of running tracks [12,13] and artificial turf football pitches [14,15]. However, despite being one of the most commonly used surfaces for sports practice worldwide, the SA, VD, and ER of treadmill surfaces remain not only unregulated but also unknown.

Treadmills are a very popular equipment for fitness training and an important resource for experienced runners, with 14% of them preferring to run on a treadmill [16]. Treadmills are also widely used in research and clinical practice for a wide variety of purposes. However, the importance of treadmill mechanical properties is often overlooked. Indeed, the main regulation concerning these machines, the European Standard EN 957-6:2010+A1:2014 [17], fails to include any reference to the assessment of the surface properties, and all it requires for testing the running area is ‘visual inspection and dimensional test’. On the other hand, scientific studies involving the use of treadmills seldom report the mechanical behavior of the surface in question. A systematic review of the biomechanical differences between treadmill and overground running concluded that mismatches in surface stiffnesses were likely the most important—but also most neglected—variable that caused differences in running biomechanics between the two modes [5]. Taken together, the aforementioned findings indicate that the mechanical properties of treadmill surfaces represent an important variable that has been largely overlooked up until now.

There is yet no standard method for assessing the mechanical properties of treadmill surfaces despite previous attempts in the field [4,18,19,20,21,22], with only one study reporting the SA, VD, and ER of one treadmill [19]. It was therefore the aim of the present study to define a reliable and sensitive test method for assessing SA, VD, and ER in treadmill surfaces. To this aim, we determined: (i) the number of tests that should be performed in a given zone to get a reliable estimate of its mechanical properties and (ii) the ability of the test method to detect meaningful differences between zones and assess the homogeneity of the surface.

## 2. Materials and Methods 

### 2.1. Sample

Two different treadmill brands (Technogym, Cesena, Italy; and Life Fitness, Rosemont, IL) were included in the study. For each brand, an old (>7 years) and a new (<1 year) treadmill model were selected such that the final sample consisted of 42 treadmills belonging to 4 different treadmill models: (a) Technogym Jog700 Excite (T-jog; *n* = 10; age ~9 years), (b) Technogym Artis Run (T-run; *n* = 12; age < 1 year), (c) LifeFitness Integrity Series 97T (L-97T; *n* = 11; age ~8 years), and (d) LifeFitness Integrity Series DX (L-DX; *n* = 9; age < 1 year). Treadmills were accessed and assessed in 4 different fitness centers in Madrid (Spain), one for each model.

### 2.2. Procedures

Characteristics of the running surfaces were assessed by measuring SA, VD, and ER using an ‘advanced artificial athlete’ (AAA) device (Deltec Equipment, Duiven, The Netherlands), in accordance with current surface testing standards [15]. The principle of the apparatus consists of a drop test through which a mass, together with a spring and a test foot attached to its bottom, is allowed to fall onto the test specimen from a certain height. The acceleration of the mass is recorded during the impact and, from these data, the SA, VD, and ER are calculated. The AAA is currently used for assessing the aforementioned three variables in artificial turf surfaces according to the regulations of the governing bodies in football, rugby, and hockey [15,23,24]. Previous studies have described the equivalence between the AAA and the ’artificial athlete’ (AA), which has been traditionally used to assess SA and VD in running tracks and other sport surfaces [25]. The AAA reports results with a sensitivity of one decimal in the three variables. Therefore, this precision was maintained to achieve the greatest accuracy in the results.

Total length (L) of treadmill surfaces was measured before testing. L was considered as the distance between the drive roller (in front of the athlete, underneath the treadmill control system) and the idler roller (behind the athlete), with the origin placed on the center of the drive roller. SA, VD and ER were assessed at 3 zones (Z1, Z2, and Z3) in each treadmill along the longitudinal axis of its surface at 1/4 L, 1/3 L, and 1/2 L, respectively (Figure 1). These zones were selected because they represent the edges of the support area (Z2, Z3) and the spot where the first contact with the surface occurs (Z1) (check the supplementary files for evidence). Three repetitions of the drop test were performed in each zone following the protocol typically used with the AAA:The apparatus was set vertically on the treadmill surface using a custom-made cart over which the AAA rested (Figure 2).The test foot was lowered smoothly onto the surface of the test specimen.A lift height of 55.00 ± 0.25 mm was set before hanging the falling mass back on the magnet, following EN 14,808 and the FIFA Handbook of Test Methods.First impact: After 30 (±5) seconds (to allow the test specimen to relax after removal of the test mass), the mass was dropped and the acceleration signal recorded. Within 10 s after the impact, lift height was checked before re-attaching the mass back on the magnet.Second impact: After 30 (±5) seconds, the mass was dropped, and the acceleration signal recorded again. Within 10 s after the impact, lift height was checked and the mass was re-attached on the magnet.Third impact: After 30 (±5) seconds, the mass was dropped, and the acceleration signal recorded.

### 2.3. Statistical Analysis

For task (i) (i.e., determination of number of tests that should be performed in a certain location to get a reliable estimate of its mechanical properties), we assessed the relationship between the results of the first impact (R1) and the criterion commonly established by international regulations (mean result of the second and third impacts, MeanR2R3). To this end, mean values of SA, VD and ER for all R1 and all MeanR2R3 were obtained for all treadmills and compared in each of the three zones using a one-way repeated-measures ANOVA. 

Once the number of repetitions that should be performed in each test location was established, we performed task (ii) (i.e., determination of the ability of the test method to detect meaningful differences between zones and assess the homogeneity of the surface). To this end, we compared again mean values of SA, VD, and ER between different locations using a one-way ANOVA. In this and the aforementioned ANOVA, the Bonferroni test was applied post-hoc for pairwise comparisons, and the mean difference and standard deviation (SD) of the mean difference were determined for each variable. The assumption of normality was visually checked using histograms and Levene’s test was used to confirm homogeneity of variance for all tests.

The agreement between the different criteria—R1 vs. MeanR2R3 for task (i), and Z1 vs. Z2, Z1 vs. Z3, and Z2 vs. Z3 for task (ii)—was assessed by means of the Bland–Altman method. The lower and upper limits of agreement were calculated as ± 1.96 times the standard deviation of the mean bias (i.e., 95% limit of agreement [LOA]). In addition, intra-class correlation coefficient (ICC, reported together with 95% confidence interval [95%CI]) was calculated and the obtained values classified as very low (<0.20), low (0.20–0.50), moderate (0.50–0.75), high (0.75–0.90), very high (0.90–0.99), and extremely high (>0.99) [26]. The ICC form used was the two-way mixed-effects, absolute agreement, single measurement [27]. Additionally, absolute consistency was quantified using the standard error of measurement (SEM), calculated with the formula SEM = (SD of the mean difference) × √(1 − ICC).

All statistical analyses were performed with SPSS software version 21.0 (SPSS Inc., Chicago, IL, USA). The significance level was set at *p* < 0.05.

## 3. Results

The descriptive data of the sample are presented in Table 1. The mean and standard deviation is reported by each of the models that make up the sample, as well as the total sample.

Table 2 shows the results obtained when comparing the criteria of using the first impact and the average value of the second and the third impacts. A significant difference was found only for VD in Zone 2. The ICC showed an extremely high agreement between both methods for all the variables barring SA in Zones 2 and 3—in which the agreement was just slightly lower, but still very high.

Bland–Altman analysis (Figure 3) showed similar behavior of the measurements regardless of the measuring range. It was only possible to appreciate a different pattern in some models of treadmills and in some specific variables, yet consistently within the limits of agreement.

As a result of the abovementioned findings, the criterion of using a single impact was considered a reproducible method. Therefore, all the results were subsequently reported by applying this criterion. Table 3 shows the comparisons in SA, VD, and ER mean values between the three zones, and Figure 4 represents the Bland–Altman analysis for each variable.

The agreement according to the ICC ranged from moderate to high, which is not enough to ensure the actual existence of a very high agreement between the different zones. Therefore, the homogeneity of the surface cannot be guaranteed, which supports the appropriateness of testing three zones to obtain an accurate representation of the overall mechanical response of the surface. Furthermore, the SEM obtained in all cases is much higher than that obtained in the analysis between impacts.

## 4. Discussion

This study aimed to define a sensitive and reliable test method for evaluating SA, VD, and ER on treadmill surfaces with the main result being that performing one single impact in three different zones with the AAA provides an accurate representation of the treadmill mechanical properties.

Various studies have warned that differences in kinematic patterns and physiological responses between treadmill and overground running might be a consequence of the different mechanical properties of the running surfaces [5,7,20,21,28,29]. Furthermore, these varying mechanical properties have also been related to risk of musculoskeletal injuries [30,31]. Thereby, to develop a sensitive method for assessing the mechanical properties of treadmill surfaces is essential not only to avoid inaccuracies in the interpretation of the results and misleading research findings, but also to ensure that running on these surfaces is performed under safe conditions.

Our results indicate that the assessment of SA, VD, and ER by one impact provides similar information to that provided when using the average of impacts two and three, as often recommended in protocols for other surfaces [13,15]. The error generated when using the results from the first impact compared to the average of the second and third impact (SA ≤ |0.1|%, VD ≤ |0.0| mm, and ER ≤ |0.2|%) is negligible. Furthermore, the uncertainty of the measurement instrument (i.e., SA = 0.4%, VD = 0.2 mm, and ER = 0.9%, as assessed in the laboratory) is higher than the mean differences between the two criteria. Therefore, if a range of acceptable values for the mechanical properties of treadmills were to be established, performing only one impact would not affect the findings because the resulting hypothetical error would not be detectable by the apparatus. The use of one single impact contributes to improving the efficiency of the test method, since the duration of the protocol is reduced. On the other hand, our results showed that testing the three zones proposed in this study, which cover the main area used during the stance phase, provides an accurate representation of the overall mechanical response of the treadmill and allows to detect differences between zones and potential damage of the surface.

Some studies have previously assessed treadmill mechanical responses using different approaches. Jones and Doust [21] assessed ER by dropping a basketball from a certain height onto the surfaces in question (treadmill and road) and measuring vertical ball rebound. More recently, other authors [4,20,22] have reported platform stiffness by placing certain weights on top of the treadmill and measuring the VD of the treadmill bed. Asmussen, et al. [18] compared the natural frequency and the damping ratio of two treadmills by performing an experimental modal analysis. Finally, Colino, et al. [19] used the AAA device to report the SA, VD, and ER of one treadmill, but they did not actually assess the suitability of their assessment method. Therefore, although the previously reported methods might have been useful for the purposes of these studies in question, they would seem inadequate for widespread use for two main reasons: a lack of representativeness of athlete–surface interaction, and/or a lack of evidence that they are sufficiently reliable and sensitive. By contrast, the method proposed in our study appears to be reliable, sensitive, time-efficient and of potential applicability, and provides a representative indication of treadmill mechanical properties.

In addition to the above, our study confirms that the mechanical properties of treadmill surfaces might substantially different from those of other overground surfaces typically used by runners. Indeed, our results show that SA, VD, and ER of some treadmills typically used in fitness centers range from 58% to 70%, from 5 to 10 mm, and from 28% to 66%, respectively. When comparing these values with the allowed ranges set by the IAAF regulation for running track surfaces [13] (i.e., 35%–50% for SA, 0.6–2.5 mm for VD and non-available for ER), our results show that even the stiffest treadmill doubles the maximum VD allowed for athletics track surfaces, and yields higher and lower values of SA and ER, respectively. These differences might be even higher when comparing treadmills to other popular running surfaces such as concrete or asphalt, which typically present SA and VD values close to 0% and 0 mm, respectively, and ER values exceeding 90% [19]. On the other hand, treadmills could better represent the mechanical properties of other surfaces such as natural grass or trails, although no reference values were found for these surfaces.

Sassi et al. [8] previously reported that an increase in SA of the surface in question is the main mechanical property responsible for a reduction in elastic energy recovery and in overall muscle-tendon efficiency, thereby resulting in greater muscle work and higher energy demands. Although they did not actually use a treadmill, these authors estimated that an increase in SA of approximately 35% will result in an increase in the cost of running of 5%, complementing previous findings that the mechanical behavior of the surface affects muscle force generation and running economy [32,33,34,35], and supporting those who later reported varying injury risk and effort perception [36,37]. In addition, in a systematic review Miller et al. [38] found that endurance running performance is generally poorer on a motorized treadmill compared to overground, which could partly be related to differences in surface stiffness. However, there is still controversy about the effect that treadmill surfaces have on athletic performance. For instance, some authors have suggested that a more compliant behavior of the treadmill surface would enhance energy return to the runner and thereby reduce the metabolic cost of running [4,22]. By contrast, some studies have found that the metabolic cost of running increased as the treadmill surface became more compliant [19,39]. In this context, the fact that SA, VD, and ER have seldom been accounted for in previous research (which has focused more on compliance or stiffness), is a potential confounder of the reported results. 

The main limitation of our study might be that mechanical devices such as the AAA, which we used here, do not completely mimic ‘natural’ human movement, and therefore do not faithfully reproduce the athlete–surface interaction [10,40,41,42,43]. However, these devices are still recognized by most international standards and sport federations’ regulations (e.g., FIFA or IAAF, among others) as being appropriate for testing and evaluating the ‘basal’ mechanical properties of sport surfaces, without direct interaction between the athlete and the surface. Thus, they represent a practical and reproducible tool for assessing the different mechanical properties of sports surfaces and for predicting surface behavior during athletic movements [40], allowing for the comparison of different surfaces and providing evidence that they meet specification standards. Another limitation of the study is the lack of available information regarding the components and structure of the surface of the different treadmill models. These aspects could potentially influence the mechanical properties of the surface and could be useful to interpret the results and explain the differences reported in the study. Future research should attempt to relate the mechanical response of the treadmill surface to the surface components and structure.

## 5. Conclusions

Our study provides a novel and reliable test protocol for assessing SA, VD, and ER on treadmill surfaces with a low margin of error. To do so, a single impact with the AAA device should be used in each of three zones of the treadmill (1/4 L, 1/3 L, and 1/2 L). If necessary, to facilitate comparability with other surfaces, SA, VD, and ER of the treadmill should be calculated as the average of the three zones. This method has potential practical applicability. Indeed, it might allow manufacturers to better assess the quality of their treadmills, and even to adapt their mechanical properties to different purposes, ensuring that running on them mimic actual overground conditions as best as possible. Researchers could also use this method when comparing treadmill and overground locomotion, since the varying mechanical properties of the treadmill and overground surfaces are to be taken into account as potential confounders. Finally, our method might help trainers and rehabilitators to gain insight into the association between treadmill training and injury risk.

## Figures and Tables

**Figure 1 sensors-20-02724-f001:**
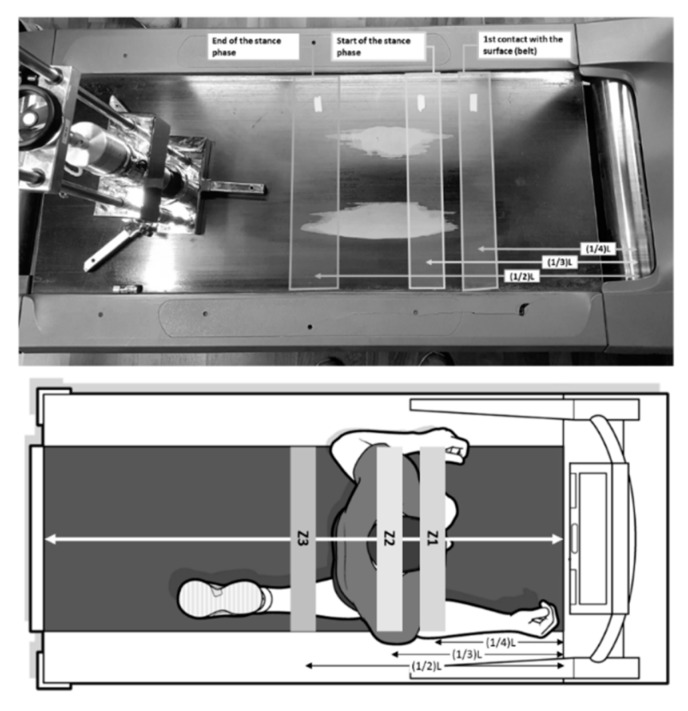
Location of the three different zones (Z) evaluated in each treadmill. Z1: (1/4)L; Z2: (1/3)L; and Z3: (1/2)L, where L is the total length of the treadmill, represented by the white line.

**Figure 2 sensors-20-02724-f002:**
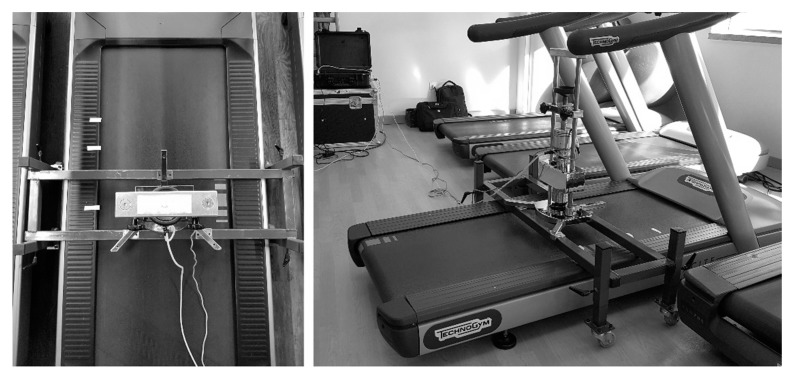
Test setup seen from above (**left**) and from the right rear side (**right**).

**Figure 3 sensors-20-02724-f003:**
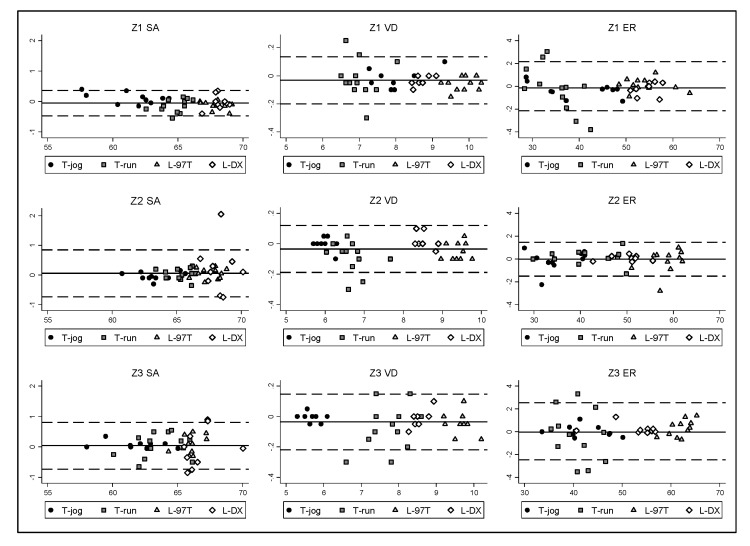
Bland–Altman analysis of the agreement between R1 and MeanR2R3. The *y*-axis represents the mean difference (R1 minus MeanR2R3) and the *x*-axis represent the average of SA (%), VD (mm), and ER (%). T-jog, T-run, L-97T, L-DX are the different treadmill models.

**Figure 4 sensors-20-02724-f004:**
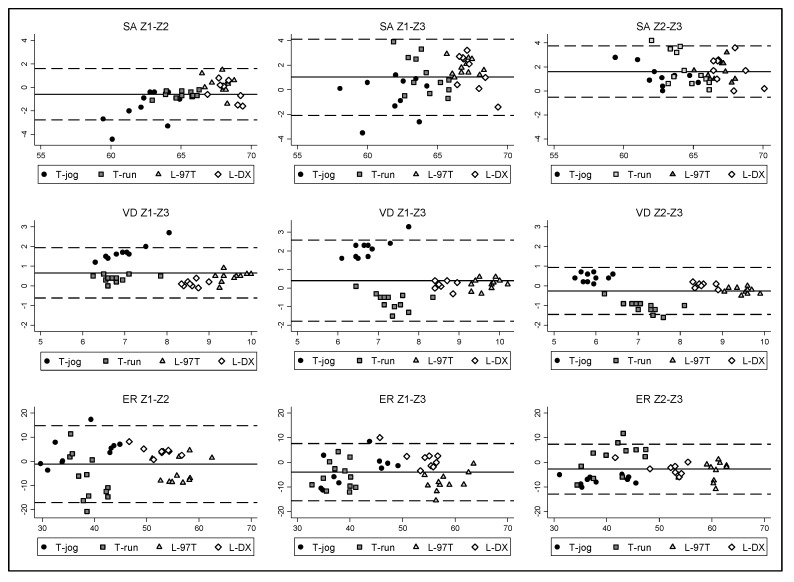
Bland–Altman analysis of the agreement between the different zones. The *x*-axis represents the mean difference between the results obtained in paired zones and the *y*-axis represents the averaged SA (%), VD (mm), and ER (%). T-jog, T-run, L-97T, L-DX are the different treadmill models.

**Table 1 sensors-20-02724-t001:** Descriptive statistics of the sample (mean ± SD).

		T-Jog	T-Run	L-97T	L-DX	Total
Zone 1	SA (%)	61.5 ± 2.3	64.8 ± 1.0	67.9 ± 0.9	68.1 ± 0.6	65.6 ± 3.0
	VD (mm)	7.8 ± 0.7	7.0 ± 0.4	9.8 ± 0.4	8.7 ± 0.2	8.3 ± 1.2
	ER (%)	39.9 ± 8.1	35.3 ± 4.6	54.1 ± 4.6	54.6 ± 2.3	45.5 ± 10.1
Zone 2	SA (%)	63.3 ± 1.5	65.3 ± 1.0	67.6 ± 0.9	68.3 ± 1.0	66.1 ± 2.2
	VD (mm)	6.1 ± 0.3	6.7 ± 0.4	9.4 ± 0.3	8.5 ± 0.2	7.6 ± 1.4
	ER (%)	35.4 ± 4.8	42.1 ± 6.8	58.5 ± 3.3	50.3 ± 3.6	46.6 ± 10.0
Zone 3	SA (%)	62.0 ± 2.1	63.5 ± 1.8	66.0 ± 0.8	66.7 ± 1.4	64.5 ± 2.4
	VD (mm)	5.7 ± 0.3	7.7 ± 0.6	9.6 ± 0.3	8.5 ± 0.2	7.9 ± 1.5
	ER (%)	42.5 ± 5.0	40.8 ± 3.9	62.0 ± 2.3	52.9 ± 5.2	49.3 ± 9.7

Abbreviations: SD, standard deviation.

**Table 2 sensors-20-02724-t002:** Comparison of R1 vs. MeanR2R3 for assessing shock absorption (SA), vertical deformation (VD), and energy restitution (ER) in each zone. ‘Mean Difference’ is R1 minus MeanR2R3.

	Mean Difference	*p*-Value	SD of the Mean Difference	ICC	95% CI		SEM
					Lower	Upper	
**Zone 1**							
SA (%)	−0.1	0.314	0.2	0.997	0.995	0.999	0.01
VD (mm)	0.0	0.057	0.1	0.997	0.994	0.998	0.00
ER (%)	0.2	1.000	1.2	0.993	0.988	0.996	0.10
**Zone 2**							
SA (%)	0.1	1.000	0.4	0.984	0.970	0.991	0.05
VD (mm)	0.0	0.020	0.1	0.998	0.996	0.999	0.00
ER (%)	0.0	1.000	0.8	0.997	0.995	0.998	0.04
**Zone 3**							
SA (%)	0.0	1.000	0.4	0.987	0.977	0.993	0.04
VD (mm)	0.0	0.052	0.1	0.998	0.995	0.999	0.00
ER (%)	0.0	1.000	1.3	0.992	0.985	0.996	0.11

* *p* < 0.05; Abbreviations: 95%CI, 95% confidence interval (for ICC); ER, energy restitution; ICC, intra-class correlation coefficient; SA, shock absorption; SD, standard deviation; SEM, standard error of measurement; VD, vertical deformation.

**Table 3 sensors-20-02724-t003:** Inter-zone comparison of shock absorption (SA), vertical deformation (VD), and energy restitution (ER). ‘Mean Difference’ is the zone on the left side minus the zone on the right.

	Mean Difference	*p*-Value	SD of the Mean Difference	ICC	95% CI		SEM
					Lower	Upper	
**SA (%)**							
Z1 vs. Z2	−0.6	0.888	1.1	0.889	0.751	0.946	0.37
Z1 vs. Z3	1.0	0.216	1.6	0.778	0.487	0.895	0.74
Z2 vs. Z3	1.6	0.015	1.1	0.728	−0.048	0.914	0.57
**VD (mm)**							
Z1 vs. Z2	0.7	0.082	0.7	0.771	0.186	0.915	0.31
Z1 vs. Z3	0.4	0.533	1.1	0.630	0.400	0.784	0.68
Z2 vs. Z3	−0.3	1.000	0.6	0.898	0.796	0.947	0.19
**ER (%)**							
Z1 vs. Z2	−1.2	1.000	8.1	0.673	0.468	0.809	4.64
Z1 vs. Z3	−4.0	0.201	5.9	0.763	0.439	0.890	2.89
Z2 vs. Z3	−2.8	0.598	5.2	0.834	0.641	0.918	2.11

Abbreviations: 95%CI, 95% confidence interval (for ICC); ER, energy restitution; ICC, intra-class correlation coefficient; SA, shock absorption; SD, standard deviation; SEM, standard error of measurement; VD, vertical deformation.

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
