# Peer review of "A Proposed Method to Assess the Mechanical Properties of Treadmill Surfaces"

_sensors, 2020, doi:10.3390/s20092724_

Round 1

Reviewer 1 Report

A Proposed Method to Assess the Mechanical Properties of Treadmill Surfaces

This study aimed to define a reliable and valid test method for assessing the Shock Absorption (SA), Vertical Deformation (VD) and Energy Return (ER) of treadmill surfaces.  Forty two treadmills were tested once in each of three zones along the treadmill belt; the test consisted of three drops of the AAA.  The results indicated that only the first drop was needed to reliably quantity the properties for a given zone.  The resulting single drop test method was demonstrated capable of detecting meaningful differences between zones.

General comments

A relevant and generally well-presented study, however I have some issues with the terminology used, specifically the use of “valid” as well as with some of conclusions that have been drawn from the data (recommendation of using the average of three locations).

Specific comments

Introduction

Paragraphs 2 and 3: Can the authors review their technical wording across these paragraphs.  For example, paragraph 2 suggests that compliance and stiffness are different properties (sand vs trampoline comparison) but later suggest the two are interchangeable.  Both sand (at least dry sand) and trampolines have high compliance (low stiffness), i.e. they deform a lot under load, the difference in their responses is due to their viscoelastic (or damping) properties relevant during the unloading phase - which are not mentioned.

The two paragraphs are also quite confusing in terms of the definitions, starting with compliance, moving to energy return, going back to compliance and then “Finally” returning to energy return.  Please also consider re-structuring.  Noting that SA and VD are both loading responses while ER is a loading and unloading response, may be an appropriate means of doing this.

Page 2, line 52: I suggest starting a new paragraph at “In order to minimise injury risk …” noting that you cannot “prevent” injury risk.

Page 2, lines 76 – 80: Make it clearer what you mean by reliable and valid.  The usual meaning of valid in scientific research relates the closeness of the values to the “truth” which is not what you have done.  Sensitivity may be a better descriptor based on what you have demonstrated.

Methods

Page 2, lines 83-89: I recommend adding a little on the running surface support structure for each treadmill group.  Notably were they designed such that you would expect consistent surface properties throughout the area of interest, noting that some treadmills are constructed to have variable stiffness along their length.

Page 3, lines 104 – 107: Can you clarify where in the medial-lateral direction you tested within each zone, e.g. was it along the centreline?  Did you investigate medial-lateral deviations in the surface properties.

Page 3, line 132: If comparing R1 vs meanR2R3 is this not a paired t-test rather than a repeated-measures ANOVA?

Results

I think it would be useful to include a summary Table of the mean and SD in SA, VD, ER for each of the four groups of treadmill in each of the three zones so the fundamental properties are clearer to the reader.  They are referenced in the Discussion and although can be generally obtained from the scales on the graphs, providing them directly would be useful.

Table 1 – Confirm the Mean Difference is R1 minus meanR2R3.

Table 1 – What was the precision of the individual measurements?  I though the AAA outputted to the nearest % in SA and ER and 0.1 mm in VD; if so, can you justify the resolution of the data presented?

Table 1 and Figure 2 – Can you check the mean differences between the Table and Figure, particularly in VD, e.g. for Zone 1 it is given as -0.03 mm while the solid horizontal line in the relevant plot looks to be at -0.3 mm?

Figure 2 – x and y descriptions are the wrong way around in the figure title

Table 2 – Again confirm the mean difference is the first zone minus second zone.

Figure 3 – Linking back to my earlier comment on the construction of the treadmill surfaces, better justify the reasons for presenting the zone comparisons as a Bland Altman, i.e. evidence that it would be expected all zones would have the same properties.

Figure 3 – It is clear from this figure that the magnitudes of some of the SDs are representing differences in the properties between the different groups of treadmills (notably VD), and don’t really represent the random errors in the data.  How does affect the analysis conducted?  Is it more relevant to complete the sensitivity analysis on a per group basis rather than across all treadmills?

Discussion

Page 7, lines 217 – 219: Again, reconsider the use of the term valid.  Also, I’m not sure why it’s clear where your three zones recommendation came from.  You tested in three zones, but why the recommendation of needing to test three zones rather than two or four or any other number?

Page 7, line 234: Where does the data on the uncertainty of the measurement instrument come from?

Page 7, line 239 – 241: I’m not sure I support making as much of the benefits of only a single drop.  Ultimately the testing time is not only the three drops but also setting up the equipment at each test location which likely takes longer than the three drops, so the time saving is a lot less than the suggested 66%.  Also, if time is a major factor then a more rigorous analysis of how many locations are really need is necessary.  Finally, other surfaces do three drops typically for good reason – i.e. the nature of the surface indicates that the surface properties change with each drop (even beyond three).  In your case it would not be expected that the surface properties would be affected by the drop process and therefore it would expected that a single drop would suffice, as you have demonstrated.

Page 8, lines 244 – 255: I’m not sure you can say that the previous methods are “inadequate” there is just a current lack of evidence that they are sufficiently reliable and sensitive.  You could also be more critical of them rather than simply listing them, e.g. ball rebound doesn’t just represent the surface response.

Page 8, lines 256 – 267: I don’t really agree with the message that emerges from this paragraph.  Overground running is performed on a wide range of surfaces of different properties – you have highlighted ones that differ to the treadmill, however natural grass and trails are very common running surfaces whose properties have quite substantial overlap with the (quite limited range of ) treadmills you tested.

Page 8, lines 268 – 283: What does the literature suggest with respect to the effects of SA on injury risk?

Page 8, lines 284 – 292: In this paragraph it is important to recognise what the AAA was designed to replicate in the context of describing the limitation.

Conclusion

Page 9, lines 297: This is the first mention that the average of the three zones should be used.  As mentioned previously it’s not clear how you arrived at the three zone recommendation (other than that’s how many you happened to test) and this recommendation should really have been in your results.

Author Response

General comments

C: A relevant and generally well-presented study, however I have some issues with the terminology used, specifically the use of “valid” as well as with some of conclusions that have been drawn from the data (recommendation of using the average of three locations).

R: Thank you very much for your comments. The authors consider that they are all very appropriate and also that they are very important to improve the quality of the article. We have tried to take account of all of them and answer in the best possible way.

Regarding the use of ‘valid’, we agree to replace it with ‘sensitive’ as the reviewer suggests below. We consider this to be a more technical and appropriate term. And as for the recommendation of using the average of three locations in the conclusion, we have rewritten this sentence to avoid misunderstanding.

Specific comments

Introduction

C: Paragraphs 2 and 3: Can the authors review their technical wording across these paragraphs.  For example, paragraph 2 suggests that compliance and stiffness are different properties (sand vs trampoline comparison) but later suggest the two are interchangeable.  Both sand (at least dry sand) and trampolines have high compliance (low stiffness), i.e. they deform a lot under load, the difference in their responses is due to their viscoelastic (or damping) properties relevant during the unloading phase - which are not mentioned.

R: Thank you very much for your contribution. Authors agree that this part was poorly explained. Regarding paragraph 2 and the sand vs trampoline comparison, our intention was not to suggest that compliance and stiffness are different properties. In fact we were trying to highlight what the Reviewer points in his/her comment, that is, that despite being both very compliant surfaces, their behaviour is very different due to their viscoelastic properties (this is what we call the ‘spring stiffness’ of the surface, following McMahon & Greene). We have eliminated some expression to avoid misunderstanding. We consider that now everything is clearer.

C: The two paragraphs are also quite confusing in terms of the definitions, starting with compliance, moving to energy return, going back to compliance and then “Finally” returning to energy return.  Please also consider re-structuring. Noting that SA and VD are both loading responses while ER is a loading and unloading response, may be an appropriate means of doing this.

R: Thank you for your comment. We have removed one sentence and rearranged the definitions better.

C: Page 2, line 52: I suggest starting a new paragraph at “In order to minimise injury risk …” noting that you cannot “prevent” injury risk.

R: Thank you for your suggestion. We have followed your recommendation and started a new paragraph where indicated. In addition, we have rewritten this paragraph to improve readers' understanding.

C: Page 2, lines 76 – 80: Make it clearer what you mean by reliable and valid.  The usual meaning of valid in scientific research relates the closeness of the values to the “truth” which is not what you have done.  Sensitivity may be a better descriptor based on what you have demonstrated.

R: Thanks for your input. Following the reviewer’s indication we have replaced the terms ‘valid’ and 'validity' by ‘sensitive’ and ‘sensitivity’ throughout the document.

Methods

C: Page 2, lines 83-89: I recommend adding a little on the running surface support structure for each treadmill group.  Notably were they designed such that you would expect consistent surface properties throughout the area of interest, noting that some treadmills are constructed to have variable stiffness along their length.

R: Thank you for this contribution. Regarding your recommendation of adding a little on the running surface support structure for each treadmill model, authors have tried before to access that information without success, since this data is not reported by manufacturers. User manuals and technical specifications of the treadmills are usually available on the web (for example, for Technogym models please see ‘https://www.technogym.com/int/download-manuals-and-documents/’), but the only information they contain about the surface structure is its dimensions. When asked by email, manufacturers did not provide this information either. To compensate that lack of information, authors have added a summary Table of the mean and SD in SA, VD, ER for each of the four groups of treadmills in each of the three zones, so at least their fundamental properties are clarified.

C: Page 3, lines 104 – 107: Can you clarify where in the medial-lateral direction you tested within each zone, e.g. was it along the centreline?  Did you investigate medial-lateral deviations in the surface properties.

R: Thank you very much for your comment. Regarding the medial-lateral direction of the tests, they were indeed performed along the centreline. This was explained in lines 103-105: "SA, VD and ER were assessed at 3 zones (Z1, Z2, and Z3) in each treadmill along the longitudinal axis of its surface at 1/4 L, 1/3 L, and 1/2 L, respectively".

As for the medial-lateral deviations in the surface properties, we did not investigate them. All treadmill surfaces analysed were symmetrical in their morphology and construction along the longitudinal axis, so we assumed that side-facing deviations from the centreline will also be symmetrical. Authors will be happy to include this as a limitation of the study or as a future line of research if the reviewer considers it important.

C: Page 3, line 132: If comparing R1 vs meanR2R3 is this not a paired t-test rather than a repeated-measures ANOVA?

R: The ANOVA was applied for operability, since it could be used at both the comparisons by repeated measures in the analysis between impacts using a repeated measures model with a single factor (Table 1 of the manuscript before review), and the differences between areas using a model of independent measurements using a single factor (Table 2 of the manuscript before review). The authors have replicated the analysis with a paired t-test and, as expected, the reported results are the same (difference in means, significance value and confidence intervals). Therefore, the use of one statistic or another in this case is indifferent, since the ANOVA is also sensitive for detecting differences, although only two groups are compared.

Results

C: I think it would be useful to include a summary Table of the mean and SD in SA, VD, ER for each of the four groups of treadmill in each of the three zones so the fundamental properties are clearer to the reader.  They are referenced in the Discussion and although can be generally obtained from the scales on the graphs, providing them directly would be useful.

R: Thank you very much for your suggestion. According to the reviewer’s contribution, authors have added a new Table to the manuscript. 

C: Table 1 – Confirm the Mean Difference is R1 minus meanR2R3.

R: Authors have added “’Mean Difference’ is R1 minus MeanR2R3” in the table title.

C: Table 1 – What was the precision of the individual measurements?  I though the AAA outputted to the nearest % in SA and ER and 0.1 mm in VD; if so, can you justify the resolution of the data presented?

R: Thank you for your contribution. Indeed, our AAA report the results to the nearest 0.1% and 0.1 mm. This information has been added to the Methods section, and all values related to the mean, mean differences and SD have been reported to the nearest 0.1.

C: Table 1 and Figure 2 – Can you check the mean differences between the Table and Figure, particularly in VD, e.g. for Zone 1 it is given as -0.03 mm while the solid horizontal line in the relevant plot looks to be at -0.3 mm?

R: Authors have checked Figure 2 and the data seem to agree with what appears in Table 1, at least in our manuscript. Regarding the example given by the reviewer, please note that the dashed line below is at -0.2 mm, so the solid horizontal line cannot be at -0.3 for being above the previous one. However, the authors have improved the quality of the graph to avoid visual misunderstandings.

C: Figure 2 – x and y descriptions are the wrong way around in the figure title

R: Thank you very much for noting this. The error has been corrected.

C: Table 2 – Again confirm the mean difference is the first zone minus second zone.

R: The following statement was included in the table title: “‘Mean Difference’ is the zone on the left side minus the zone on the right”

C: Figure 3 – Linking back to my earlier comment on the construction of the treadmill surfaces, better justify the reasons for presenting the zone comparisons as a Bland Altman, i.e. evidence that it would be expected all zones would have the same properties.

R: Although any sports surface is designed to be uniform throughout its areas, various zones must be evaluated to ensure consistency across the entire surface and get more rigorous results. Most manufacturers develop sport surfaces to be homogeneous (i.e. football pitches, athletics tracks and most treadmills), but they may not meet their objective. For this reason, one of the quality requirements of any surface is to demonstrate that there is a minor variation across different zones, normally limiting variations to 10% with respect to the average of all zones. Therefore, 3 test locations were defined following the philosophy of the regulations, and the agreement was also evaluated in the results of those areas to obtain the maximum information.

C: Figure 3 – It is clear from this figure that the magnitudes of some of the SDs are representing differences in the properties between the different groups of treadmills (notably VD), and don’t really represent the random errors in the data.  How does affect the analysis conducted?  Is it more relevant to complete the sensitivity analysis on a per group basis rather than across all treadmills?

R: Interesting consideration, thank you very much for your comment. Although figures were divided by groups, authors considered appropriate to analyse data as a whole, since the aim of the study was to establish a reliable and sensitive test method for assessing all treadmill surfaces, regardless of its brand or model. For this reason, the authors consider that the established model applies across all treadmills, despite being perhaps too conservative in certain cases.

Discussion

C: Page 7, lines 217 – 219: Again, reconsider the use of the term valid.  Also, I’m not sure why it’s clear where your three zones recommendation came from.  You tested in three zones, but why the recommendation of needing to test three zones rather than two or four or any other number?

R: Thank you for your comment. The use of the term valid has been corrected. As for the other issue, please consider the following:

To establish the protocol, we needed to know if it was different or not to carry out trials in 1 area, in 2 or more. Thus, we initially selected these 3 zones because they represent (approximately) the edges of the support area (Z2, Z3) and the spot where the first part of the contact with the surface seems to occur (Z1) (please check new Figure 1 for evidences). By doing this, our aim was to test the part of the surface with which athletes interact, and these three zones were considered representative in the beginning. Then, we investigate differences between them to reduce the number of zones as much as possible. If we had not identified differences between them, we would have discussed that testing only one is enough. But the existence of differences led us to consider that the three of them should be tested.

In this sense, it should be noted that we recommend assessing 3 zones in a relatively small surface. In contrast, only 6 zones must be evaluated in a football pitch according to EN standards, or 19 according to FIFA standards. By using 3 zones in the treadmills, we guarantee that the entire surface used by the athlete is analyzed, and we ensure the reliability of the established protocol.

C: Page 7, line 234: Where does the data on the uncertainty of the measurement instrument come from?

R: Thank you for your comment. These data come from an external laboratory accredited to calibrate the test instrument under the corresponding regulations. Thus, the authors guarantee that the values reported are in accordance with European standards for each type of test. Authors will be happy to attach these calibration certificates as supplementary files if the reviewer wants to analyze them.

C: Page 7, line 239 – 241: I’m not sure I support making as much of the benefits of only a single drop.  Ultimately the testing time is not only the three drops but also setting up the equipment at each test location which likely takes longer than the three drops, so the time saving is a lot less than the suggested 66%.  Also, if time is a major factor then a more rigorous analysis of how many locations are really need is necessary.  Finally, other surfaces do three drops typically for good reason – i.e. the nature of the surface indicates that the surface properties change with each drop (even beyond three).  In your case it would not be expected that the surface properties would be affected by the drop process and therefore it would expected that a single drop would suffice, as you have demonstrated.

R: Thank you very much for this comment. Authors fully agree with this observation. Since no previously established protocol was available for this type of surfaces, our initial intention was to respect the existing protocols for other sport surfaces, and then investigate whether that protocol could be optimized in the case of treadmills. We share the opinion that the reduction of time is anecdotical and not a major factor in this study, so we have modified that phrase to avoid confusion.

C: Page 8, lines 244 – 255: I’m not sure you can say that the previous methods are “inadequate” there is just a current lack of evidence that they are sufficiently reliable and sensitive.  You could also be more critical of them rather than simply listing them, e.g. ball rebound doesn’t just represent the surface response.

R: Thank you for this contribution. Authors have rewritten this statement, including a criticism of the lack of representativeness of some previously used methods.

C: Page 8, lines 256 – 267: I don’t really agree with the message that emerges from this paragraph.  Overground running is performed on a wide range of surfaces of different properties – you have highlighted ones that differ to the treadmill, however natural grass and trails are very common running surfaces whose properties have quite substantial overlap with the (quite limited range of) treadmills you tested.

R: Thank you very much for your comment. We have softened the message emerging from this paragraph by re-writing some parts and adding this sentence at the end: “On the other hand, treadmills could better represent the mechanical properties of other surfaces such as natural grass or trails, although no reference values were found for these surfaces”. Authors also agree that the range of treadmills tested here might be scarce, and we are now working to try and get access to a greater number of them, including treadmills designed for research and clinical purposes. This would be the purpose of a future research.

C: Page 8, lines 268 – 283: What does the literature suggest with respect to the effects of SA on injury risk?

R: Thank you for your comment. After reviewing the literature, there did not seem to be much evidence linking SA directly with injury risk. Instead of SA, surface stiffness (which would be better represented by VD instead of SA) has sometimes been related to other variables such as muscle activity, foot pressure or kinematic patterns, which in turn have been linked to injury risk. Since the evidence is scarce and often conflicting (please check ‘van Hooren et al.’ for more details), authors decided not to talk in depth about this topic (implications of the mechanical properties of the treadmills) as it moves away from the main objective of the study (to design a test protocol to assess them in an efficient way which is also consistent with current standards for other sport surfaces). Without prejudice of the previous, we have added some bibliographic information on this topic, including two papers that assessed the same mechanical properties (SA, VD and ER) and linked them somehow to injury risk and effort perception. Thus, this paragraph is now more complete. Thank you very much for this contribution.

C: Page 8, lines 284 – 292: In this paragraph it is important to recognise what the AAA was designed to replicate in the context of describing the limitation.

R: Thanks for noting this. We have added this information as suggested by the reviewer.

Conclusion

C: Page 9, lines 297: This is the first mention that the average of the three zones should be used.  As mentioned previously it’s not clear how you arrived at the three zone recommendation (other than that’s how many you happened to test) and this recommendation should really have been in your results.

R: Thank you very much for this contribution. Authors have nuanced the reference to the average of the three zones being used. As for the last part of the comment, authors have explained before why these 3 zones were chosen and the protocol to determine them, and we consider that this has now been sufficiently described in the methodology section. Furthermore, authors have modified Figure 1 to clarify this question.

Reviewer 2 Report

The work is interesting and novel. However, the authors need to do major revisions and resubmit the manuscript based on following comments.

1 - The abstract is waffle, please rewrite it again

2 - The introduction doesnt pitch the idea of work being presented. Please revised it 

3 - Figure 1 need to be replaced with real picture of the treadmill.

4 - The resolutoin of figure 2 is too low. please improve.

5 - This paper lacks state of the art work. Please include following work to improve quality of references.Since these works are related to sensors and related to work presented in this paper.

      a . Shah, S.A., Fan, D., Ren, A. et al. Seizure episodes detection via smart medical sensing system. J Ambient Intell Human Comput (2018).

       b .F. Fioranelli, J. Le Kernec and S. A. Shah, "Radar for Health Care: Recognizing Human Activities and Monitoring Vital Signs," in IEEE Potentials, vol. 38, no. 4, pp. 16-23, July-Aug. 2019. 

        c. D. Haider et al., “An efficient monitoring of eclamptic seizures in wireless sensors networks,” Comput. Electr. Eng., vol. 75, pp. 16–30, 2019.

         d. Shah, SA, Yang, X, Abbasi, QH. Cognitive health care system and its application in pill‐rolling assessment. Int J Numer Model. 2019;e2632. https://doi.org/10.1002/jnm.2632

Author Response

The work is interesting and novel. However, the authors need to do major revisions and resubmit the manuscript based on following comments.

1 - The abstract is waffle, please rewrite it again

Thank you for your comment. We have rewritten some parts of the abstract following your advice.

2 - The introduction doesnt pitch the idea of work being presented. Please revised it 

Thank you for your contribution. Authors have clarified certain sections in the introduction. We consider that the main idea of this study is now presented more clearly.

3 - Figure 1 need to be replaced with real picture of the treadmill.

Thank you very much for your suggestion. We have adapted Figure 1 following your recommendations.

4 - The resolutoin of figure 2 is too low. please improve.

Thank you for your contribution. Authors have reviewed the quality of all Figures in the manuscript. We believe that their resolution is now high enough.

5 - This paper lacks state of the art work. Please include following work to improve quality of references.Since these works are related to sensors and related to work presented in this paper.

Thank you very much for this contribution and your suggestions. We understand your concern. Authors have added 2 new references related to the study topic. However, we have decided not to include these references as they are not related to the objective of the present study. We appreciate your recommendations and take note for future research.

Reviewer 3 Report

General comments

I enjoyed reading this paper investigating the mechanical properties of various treadmills. The study has relevance both to industry and research. Overall it is very well written and clear in the presentation of findings. My main concern is with the inclusion of the “smallest worthwhile change” analysis. This statistical approach has been largely discredited and it doesn’t add anything to the current study, so I suggest it be removed. This will require changes to the Results and Discussion sections. Otherwise, I have made some suggestions where I think there could be improvement before proceeding to publication.

Specific comments

Abstract

Lines 19-23: Please change wording to reflect change in statistical analysis used to support these statements.

Introduction

What constitutes the mechanical shock absorption properties of a treadmill? The shocks, belt tightness, etc.? This warrants discussion in the Introduction and also the Discussion.

P.2, Line 52: “…prevent injury risk...” Please change to “reduce injury risk.”

P.2, Line 56: The IAAF is now known as “World Athletics.”

P.2, Line 73: “…up till now.” Please change to “…up until now.”

Methods

P.2, Lines 83-89: Was any effort made to determine how heavily used the treadmills were? Or was there any control for this?

P.3, Line 94: Was the test foot wearing a shoe? If so, how were the mechanical properties of the shoe and/or foot taken into account?

P.3, Line 94-95: It would be useful to have a figure (photo or diagram) that shows the test setup.

P.3, Line 107: No supplementary files were provided.

P.3, Line 109: Was the apparatus set so that the test foot fell in the midline of the belt or to the side as a foot would normally fall on a treadmill when running?

P.4, Lines 152-155: I strongly recommend removing this analysis (SWC) from the manuscript. As a statistical tool, it has largely been discredited for magnifying results. The statistical analysis you have already done should be enough to support your results.

Results

Why not report overall results of SA, VD, and ER (Mean+/- SD) by treadmill model/age? You do discuss these findings in the Discussion (P.8, Lines 256-267).

Please make changes to Results section based on removal of SWC analysis

Discussion

It appears that the treadmill was not running during the testing procedure. How might this have influenced the results? Is there any change in belt tension as a result of the treadmill belt moving?

P.7, Lines 231-235: See previous concerns over the use of the SWC analysis. The second point made here (error of the measurement instrument) is a strong enough rationale that the first statement regarding SWC is not needed.

P.8, Line 281: Citations 19 and 38 not square-bracketed.

Tables & Figures

Please provide a table showing the results of the SA, VD, and ER of the various treadmill models.

Please make changes to Tables section based on removal of SWC analysis

Please add a figure (photo or diagram) that shows the test setup.

Author Response

General comments

I enjoyed reading this paper investigating the mechanical properties of various treadmills. The study has relevance both to industry and research. Overall it is very well written and clear in the presentation of findings. My main concern is with the inclusion of the “smallest worthwhile change” analysis. This statistical approach has been largely discredited and it doesn’t add anything to the current study, so I suggest it be removed. This will require changes to the Results and Discussion sections. Otherwise, I have made some suggestions where I think there could be improvement before proceeding to publication.

Thank you

Specific comments

Abstract

C: Lines 19-23: Please change wording to reflect change in statistical analysis used to support these statements.

R: Done. Thank you very much for this contribution.

Introduction

C: What constitutes the mechanical shock absorption properties of a treadmill? The shocks, belt tightness, etc.? This warrants discussion in the Introduction and also the Discussion.

R: Thank you very much for your comment. Authors did share the Reviewer’s concern in this sense, not only with regard to the mechanical shock absorption (SA) but also to the ability to deform under load (VD) and the energy restitution (ER). For that reason, authors tried without success to gain knowledge about the running surface support structure for each treadmill model used in the study. However, this information is not reported by manufacturers. User manuals and technical specifications of the treadmills are usually available on the web (for example, for Technogym models please see ‘https://www.technogym.com/int/download-manuals-and-documents/’), but the only information they contain about the surface structure is its dimensions. When asked by email, manufacturers did not share this information either.

Authors consider that, in order to respond to the reviewer’s question (or at least to hypothesize about it), we would need some data about the treadmills’ materials and construction that we do not have, so we do not consider appropriate to theorize about it because of this lack of reliable information. Nevertheless, the purpose of the present study was to design a test method to evaluate the mechanical properties of treadmill surfaces, independently of which factors affect those results. Investigating these factors may be the purpose of future research but, to do so, a test method to measure the mechanical properties of treadmill surfaces should be previously available.

Authors will be happy to acknowledge such lack of information as a limitation of the study and to include that as a future line of research, if the reviewer considers it important.

C: P.2, Line 52: “…prevent injury risk...” Please change to “reduce injury risk.”

R: Done. Thank you for your contribution.

P.2, Line 56: The IAAF is now known as “World Athletics”

C: Thanks, this has been clarified.

R: P.2, Line 73: “…up till now.” Please change to “…up until now.”

Done. Thank you.

Methods

C: P.2, Lines 83-89: Was any effort made to determine how heavily used the treadmills were? Or was there any control for this?

R: Thank you for your comment. Just like in the case of the treadmills’ structure commented before, authors tried to answer that question before without success. When asked about the kilometers completed or the hours of use, the managers of the fitness centres told us that this information could only be had, at most, by the personnel who carried out the maintenance of the treadmills. In all cases, these personnel did not belong to the organization itself. In three of them they belonged to certain subcontracted companies, and in the other one it was the manufacturer itself, since the treadmills had been recently purchased and maintenance during the first years was included. When contacting the subcontracted companies, we were told that they did not have this information, and that it could only be known by the manufacturer itself, if at all possible. Therefore, the only information that may help to get an idea of how heavily used were the treadmills is their age. In this sense, we tried our best to include old and new treadmills to improve the representativeness of the study.

C: P.3, Line 94: Was the test foot wearing a shoe? If so, how were the mechanical properties of the shoe and/or foot taken into account?

R: Thank you for your comment. No, the test foot was not wearing a shoe. The evaluation of SA, VD and ER was carried out imitating the protocols established both at European level (with EN standards) and at federative level (with World Athletics, FIFA, World Rugby… standards). These regulations describe the mechanical apparatus that has to be used for the evaluation of the mechanical properties of the surfaces, and none of them includes adding a shoe to the test foot. That way they make sure to evaluate the SA, VD and ER of the surface itself, without considering the ‘athlete’ constraint in the athlete-surface interaction.

C: P.3, Line 94-95: It would be useful to have a figure (photo or diagram) that shows the test setup.

R: Thank you for your suggestion. The new Figure 2 shows the test setup from above and from a lateral view.

C: P.3, Line 107: No supplementary files were provided.

R: Thank you very much for noting this. Authors have solved this problem by removing that statement from the manuscript and adding the evidence to Figure 1.

C: P.3, Line 109: Was the apparatus set so that the test foot fell in the midline of the belt or to the side as a foot would normally fall on a treadmill when running?

R: Thank you for your comment. The test foot did indeed fall in the centreline. This was explained in lines 103-105: "SA, VD and ER were assessed at 3 zones (Z1, Z2, and Z3) in each treadmill along the longitudinal axis of its surface at 1/4 L, 1/3 L, and 1/2 L, respectively".

Authors did not investigate side-facing deviations from the centreline for three main reasons: the fact that different athletes may land their feet at different widths; the fact that medial-lateral deviations are supposed to be symmetrical along the longitudinal axis of the treadmills; and also that the complexity of the test protocol would increase considerably. Authors will be happy to include this as a limitation of the study or as a future line of research if the reviewer considers it important.

C: P.4, Lines 152-155: I strongly recommend removing this analysis (SWC) from the manuscript. As a statistical tool, it has largely been discredited for magnifying results. The statistical analysis you have already done should be enough to support your results.

R: Done. Thank you.

Results

C: Why not report overall results of SA, VD, and ER (Mean+/- SD) by treadmill model/age? You do discuss these findings in the Discussion (P.8, Lines 256-267).

R: A new table (Table 1) has been included in the Results section.

C: Please make changes to Results section based on removal of SWC analysis

R: Done. Thank you.

Discussion

C: It appears that the treadmill was not running during the testing procedure. How might this have influenced the results? Is there any change in belt tension as a result of the treadmill belt moving?

R: Thank you for your comment. Indeed, as for any other sport surface, the mechanical properties of the treadmills were tested with no relative speed between the surface and the test apparatus. The current test apparatus recognized in the standards do not allow to do so in a different way. As for the reviewer’s concern, authors hypothesize that belt tension should not change as a result of the treadmill belt moving since the wheelbase does not change in any case. Therefore, this lack of movement should not influence the results.

C: P.7, Lines 231-235: See previous concerns over the use of the SWC analysis. The second point made here (error of the measurement instrument) is a strong enough rationale that the first statement regarding SWC is not needed.

R: Thank you very much for your contribution. We have removed everything related to SWC following your recommendations.

C: P.8, Line 281: Citations 19 and 38 not square-bracketed.

R: Thanks, we have solved this issue.

Tables & Figures

C: Please provide a table showing the results of the SA, VD, and ER of the various treadmill models.

R: A new table (Table 1) has been included in Results section.

C: Please make changes to Tables section based on removal of SWC analysis

R: Done.

C: Please add a figure (photo or diagram) that shows the test setup.

R: Done. Figure 2 shows now the test setup.

Round 2

Reviewer 2 Report

The authors have addressed all my comments. I would recommend this manuscript for publication.

Author Response

Dear reviewer, 

Thank you so much for your comments and your help to improve de the quality of this paper.

All the best,

Reviewer 3 Report

A proposed method to assess the mechanical properties of treadmill surfaces

General comments

Thank you for addressing my concerns and comments regarding the first version of this manuscript. The authors have done an excellent job of addressing these concerns and I believe the paper is stronger.

Specific comments

Thank you for explaining your efforts to determine what constitutes mechanical shock absorption in commercial treadmills. Please add this as a limitation.

Author Response

Thank you very much for your contribution. Authors have added the following sentence at the end of the Discussion section: “Another limitation of the study is the lack of available information regarding the components and structure of the surface of the different treadmill models. These aspects could potentially influence the mechanical properties of the surface and could be useful to interpret the results and explain the differences reported in the study. Future research should attempt to relate the mechanical response of the treadmill surface to the surface components and structure.”